# Synthesis of Sugar and Nucleoside Analogs and Evaluation of Their Anticancer and Analgesic Potentials

**DOI:** 10.3390/molecules27113499

**Published:** 2022-05-29

**Authors:** Fahad Hussain, Fahad Imtiaz Rahman, Poushali Saha, Atsushi Mikami, Takashi Osawa, Satoshi Obika, S. M. Abdur Rahman

**Affiliations:** 1Department of Clinical Pharmacy and Pharmacology, Faculty of Pharmacy, University of Dhaka, Dhaka 1000, Bangladesh; phar.fahad@gmail.com (F.H.); fahad@du.ac.bd (F.I.R.); poushali@du.ac.bd (P.S.); 2Graduate School of Pharmaceutical Sciences, Osaka University, 1-6 Yamadaoka, Osaka 565-0871, Japan; mikami-a@phs.osaka-u.ac.jp (A.M.); oosawa-t@phs.osaka-u.ac.jp (T.O.); obika@phs.osaka-u.ac.jp (S.O.)

**Keywords:** synthesis, sugar derivatives, nucleoside analogs, anticancer, analgesic, molecular docking

## Abstract

Chemical modification of sugars and nucleosides has a long history of producing compounds with improved selectivity and efficacy. In this study, several modified sugars (**2**–**3**) and ribonucleoside analogs (**4**–**8**) have been synthesized from α-d-glucose in a total of 21 steps. The compounds were tested for peripheral anti-nociceptive characteristics in the acetic acid-induced writhing assay in mice, where compounds **2**, **7**, and **8** showed a significant reduction in the number of writhes by 56%, 62%, and 63%, respectively. The compounds were also tested for their cytotoxic potential against human HeLa cell line via trypan blue dye exclusion test followed by cell counting kit-8 (CCK-8) assay. Compound **6** demonstrated significant cytotoxic activity with an IC_50_ value of 54 µg/mL. Molecular docking simulations revealed that compounds **2**, **7**, and **8** had a comparable binding affinity to cyclooxygenase-1 (COX-1) and cyclooxygenase-2 (COX-2) enzymes. Additionally, the bridged nucleoside analogs **7** and **8** potently inhibited adenosine kinase enzyme as well, which indicates an alternate mechanistic pathway behind their anti-nociceptive action. Cytotoxic compound **6** demonstrated strong docking with cancer drug targets human cytidine deaminase, proto-oncogene tyrosine-protein kinase Src, human thymidine kinase 1, human thymidylate synthase, and human adenosine deaminase 2. This is the first ever reporting of the synthesis and analgesic property of compound **8** and the cytotoxic potential of compound **6**.

## 1. Introduction

With new diseases and threats emerging every other day, the importance of discovering novel molecules with pharmacological potential is now greater than ever before. Carbohydrate-based drugs have always drawn huge attention in synthetic chemistry due to their unparallel physiological recognition capacity, huge possibilities of structural variation, and diverse pharmacological potential in the treatment of different diseases [1]. In 2019, the FDA approved 48 drugs, several of them being designed based on carbohydrate scaffolds [2]. The molecular target recognition capacity of these drugs was improved by making structural changes based on structure–activity relationship (SAR) studies. Among carbohydrate-based drugs, nucleoside analogs have been in clinical use for decades [3,4,5] and some of them are even being considered therapeutic strategies in the fight against the SARS-CoV-2 pandemic outbreak [6,7]. Nucleosides show a great point of initiation for drug design because of their essential role in various important biological processes and also because of their distinguishable role as the constitutional elements in DNA and RNA synthesis [8,9].

There are more than twenty-five FDA-certified nucleoside and nucleotide analogs being utilized to target specific tumors, factoring for a significant proportion of the existing molecular cache for treating malignancy [10]. In addition, the emergence of newer cytotoxic nucleoside analogs as well as continuous improvement of the currently available analogs will contribute greatly to the search for a cure for cancer [11]. There are two reasons behind the increasing significance of nucleoside and nucleotide analogs as anticancer substances. One is the emergence of new derivatives with wide usefulness to different forms of cancer. The other reason is the more in-depth insight into their metabolic pathways, which allow their use against different cancers to be more effective [12]. The promise of nucleoside analogs in cancer-treating therapies justifies intensive ongoing research in this area and shows nucleoside frameworks could be used as lead compounds for developing new anticancer drugs in the future.

Though there is no approved sugar and nucleoside derivative available for producing analgesia, the scientific literature has evidence of the antinociceptive potential of various nucleoside analogs. Toshiyuki et al. reported the antinociceptive effect of N^3^-phenacyluridine by acetic acid-induced abdominal constriction [13]. Inspired by this study, Shimizu et al. further studied 78 N^3^-substituted derivatives of uridine, thymidine, deoxyuridine, etc., and found three new compounds with superior (five-fold) pain alleviating potential than the compound Toshiyuki et al. reported first [14]. Several other recent studies have reported the anti-nociceptive effect of sugar derivatives synthesized from α-D-ribofuranose which further justifies their research as potential analgesics [15,16].

Computer-aided drug design is now widely applied in the drug discovery process to make it less time-consuming and more cost-efficient. In silico molecular docking, strategies can play a key role in overcoming challenges in the formulation and drug development [17]. Determining enzyme peptide interactions and analyzing the most stable drug inclusion complex can play a key role in designing promising derivatives from lead compounds. While there are still several limitations and challenges in the docking methodology [18], it is still considered a vital tool in lead optimization and can provide a molecular mechanism behind the pharmacological actions demonstrated by lead compounds.

In view of the above-mentioned cytotoxic and anti-nociceptive potential of sugar derivatives and nucleoside analogs, we planned to synthesize some sugar and nucleoside derivatives starting from natural sugar. We have synthesized several sugar derivatives and nucleoside analogs, some of which are bridged nucleosides [19]. In vivo anti-nociceptive property in acetic acid-induced nociception of Swiss albino mice and in vitro cytotoxic potential of the compounds against HeLa cancer cells were evaluated. Molecular docking simulations were also performed to provide a mechanistic insight into the pharmacological actions demonstrated by the compounds.

## 2. Results

### 2.1. Chemistry

The synthesis of a number of sugar and nucleoside analogs were carried out from α-d-glucose (**1**) as depicted in Figure 1. Structural modification of the starting sugar was performed in several step-wise processes. Diacetal (**2**) was synthesized from α-d-glucose in three steps using known procedures [20]. First, the ketalization of α-d-glucose using CuSO_4_ and sulfuric acid in acetone was performed according to the method of Kawsar et al. [21]. For changing the stereochemistry of the C3 hydroxyl group, initially, oxidation of C3-hydroxyl was accomplished using TEMPO as a catalyst which works well for sterically hindered secondary alcohols [22] followed by reduction with NaBH_4_ [23]. Compound **3** was synthesized from compound **2** in a total of four steps as described in the Appendix A. First, the benzylation of the C3 hydroxyl group in compound **2** was conducted according to the reported literature [24]. Then, the 5,6-*O*-isopropylidene group was removed by acid-mediated cleavage followed by periodate oxidation of vicinal 5,6-diol, and finally, aldol condensation was performed to produce compound **3** in excellent yield according to the reported method of Sharma et al. [25].

Compound **3** was used for the synthesis of various nucleoside derivatives. Nucleoside derivative **4** was obtained in three steps from compound **3**. Selective benzylation of the C5 hydroxyl group [26], followed by deprotection of the 1,2-*O*-isopropylidene ring, acetylation [27], and finally nucleosidation by silylated thymine base produced nucleoside derivative **4** in good yield. Subsequent deacetylation furnished compound **5** in excellent yield [28]. Compound **6** was also prepared from compound **3** by a similar procedure except a tert-butyldiphenylsilyl group was introduced instead of the acetyl group. Compound **3** was also transformed to the bridged nucleoside analog **7** by a series of six reactions, which began again with the selective benzylation of the C5 hydroxyl group, followed by tosylation. Afterward, deprotection of the 1,2-*O*-isopropylidene ring followed by acetylation and then nucleosidation by silylated thymine base produced a nucleoside derivative [9]. Finally, a base-mediated cyclization to form a bridged nucleoside [29] and debenzylation produced bridged nucleoside **7** in good yields [28]. Esterification of the primary alcohol with isobutyric anhydride afforded the new bridged nucleoside **8** in excellent yield (Figure 1).

### 2.2. Biological Study

#### 2.2.1. Peripheral Analgesic Activity in Mice Model

The peripheral anti-nociceptive activity of the synthesized compounds was tested at two different doses (25 mg/kg and 50 mg/kg) and compared with negative control (0.9% NaCl saline) and positive control aceclofenac (25 mg/kg). At 50 mg/kg dose compound **2**, **7** and **8** significantly inhibited writhing by 55.56 ± 7.66% (*p* < 0.05), 61.73 ± 6.59% (*p* < 0.01) and 62.96 ± 8.05% (*p* < 0.05), respectively, compared to the negative control group (Figure 1), whereas aceclofenac inhibited writhing by 82.72 ± 5.31% (*p* < 0.001). None of the other compounds showed any significant reduction in writhing percentage as shown in Figure 1.

#### 2.2.2. Trypan Blue Dye Exclusion Test for Cytotoxic Activity

The preliminary cytotoxic potential of the synthesized compounds was tested using the trypan blue dye exclusion test. The percentage of viable cells after administration of the synthesized compounds was determined in the trypan blue dye exclusion test and has been summarized in Table 1. Among the synthesized compounds, **5** and **6** exhibited the highest rupture of cells with a viable cell count of 40–50% and less than 5%, respectively, at 500 µg/mL concentration (Figure 2). The other compounds did not show any significant rupture which gave us a preliminary idea about their cytotoxic activities.

#### 2.2.3. CCK-8 Cell Viability Assay for Cytotoxic Activity

Based on the qualitative assessment of cytotoxicity in the trypan blue dye exclusion test, compound **5** and compound **6** were further tested to determine their IC_50_ concentration using the CCK-8 cell viability assay. Compound **5** and compound **6** were administered on HeLa cells at doses of 15.62, 31.25, 62.5, 125, and 250 µg/mL, respectively, and absorbance was measured at 490 nm using a spectrophotometer to measure the viable cell count. Compound **6** demonstrated significant growth inhibition with an IC_50_ value of 54 µg/mL (76.4 µM), whereas compound **5** was not cytotoxic even at the maximum administered dose of 250 µg/mL (Table 2).

### 2.3. Molecular Docking Simulation

#### 2.3.1. Docking Simulation against Target Proteins of Analgesic and Cytotoxic Activity

In silico molecular docking simulations were conducted on the pharmacologically active compounds to understand their mechanistic pathway of activities. Docking scores against target proteins, which is the energy (kcal/mol) required to bind with the active sites of the proteins, have been determined to measure the target specificity of the active compounds. Ligand–protein interactions have also been analyzed to predict which bonds helped stabilize the docking with active sites of the proteins.

For compounds **2**, **7,** and **8,** which demonstrated analgesic activity in the peripheral anti-nociceptive mice model, docking simulations were performed against potential targets cyclooxygenase-1, cyclooxygenase-2, and human adenosine kinase, and the results have been summarized in Table 3.

For the cytotoxic compound **6**, docking simulations were performed against human cytidine deaminase, proto-oncogene tyrosine-protein kinase Src, human thymidine kinase 1, human thymidylate synthase, and human adenosine deaminase 2, which has been summarized in Table 4.

#### 2.3.2. Inhibition of Cyclooxygenase-1 and Cyclooxygenase-2

Against the active site of COX-1, compounds **2**, **7**, and **8** exhibited good binding affinity with docking scores of −6.9 kcal/mol, −6.1 kcal/mol, and −6.8 kcal/mol, respectively (Table 3). Graphical representation of their interactions revealed that compound **2** forms strong carbon–hydrogen bonds with ILE523, GLY526, and ALA527 residues in chain B of the COX-1 enzyme. Compound **7** forms strong hydrogen bonds with HIS90, GLY354, and SER516 residues in the active site of COX-1. Interestingly, aceclofenac also forms a strong hydrogen bond with HIS90 residue. Compound **8**′s docking to the active site of COX-1 is stabilized by hydrogen bonds formed with GLN351, PHE580, and HIS581 residues. Apart from these strong hydrogen bonds, the compounds also had several hydrophobic interactions with the active site in chain B of the COX-1 enzyme, which has been presented in Figure 3.

Against cyclooxygenase-2 enzyme, compounds **2**, **7,** and **8** demonstrated strong docking scores, which reflects the significant analgesic activity they exhibited during the in vivo assay. Compound **2** exhibited the strongest binding affinity towards the COX-2 enzyme (−7.1 kcal/mol), closely followed by compounds **7** (−7.0 kcal/mol) and **8** (−6.8 kcal/mol) (Table 3). The strong docking of compound **2** could be attributed to the five hydrogen bonds formed with the ARG120, TYR355, VAL523, GLY526, and ALA527 residues in the active site of the COX-2 enzyme, which resembles the interaction of aceclofenac with the VAL523 and ALA527 residues. Active hydrogen bond binding sites of compound **7** with COX-2 were SER353, GLY526, and ALA527 residues. For compound **8**, strong hydrogen bonds were formed with ARG120, TYR355, and ALA527 residues. All the interactions of protein–ligand, including both hydrogen bond and hydrophobic interactions, have been presented in Figure 4.

#### 2.3.3. Inhibition of Human Adenosine Kinase

Compounds **2**, **7**, and **8** exhibited a strong binding affinity towards human adenosine kinase in comparison to its potent inhibitor 5-ethynyl-7-(beta-D-ribofuranosyl)-7H-pyrrolo [2,3-d]pyrimidin-4-amine. Among these compounds, bridged nucleoside analogs **7** and **8** showed stronger docking scores of −7.9 kcal/mol and −7.4 kcal/mol, respectively, followed by the −6.5 kcal/mol docking score of the sugar derivative **2** (Table 3). Graphical representation of the binding interactions revealed that compound **7** formed strong hydrogen bonds with ASP18, SER65, GLY64, and ASP300 residues of the A chain of human adenosine kinase. Interestingly, the standard ligand used in our study also formed hydrogen bonds with these same residues as well. Compound **8** formed a total of seven hydrogen bonds with GLN38, GLY64, SER65, ASN68, ARG132, ASN296, and ASP300 residues in the active site of our target protein, whereas compound **2** formed stable hydrogen bonds with ASP18, SER65, ASN68, GLY297, and ASP300 residues. All the interactions of protein–ligand, including both hydrogen bond and hydrophobic interactions, have been presented in Figure 5.

#### 2.3.4. Inhibition of Human Cytidine Deaminase

Compound **6** demonstrated a stronger binding affinity to the active site of human cytidine deaminase (PDB: 1MQ0) with a docking score of −7.1 kcal/mol compared to the −6.8 kcal/mol exhibited by the enzyme’s potent inhibitor 1-beta-ribofuranosyl-1,3-diazepinone (Table 4). Graphical presentation of the binding interactions revealed that this strong inhibition could be attributed to the stable hydrogen bonds formed with ASN54, PHE95, and SER97 residues in the active site of human cytidine deaminase. Apart from these hydrogen bonds, strong pi-anion interaction with ASP94 residue and other hydrophobic interactions with PHE36, VAL38, ALA58, CYS65, and MET91 residues also contributed to the high docking score of the compound **6**. Figure 6 compares all the binding interactions of compound **6** with potent inhibitor 1-beta-ribofuranosyl-1,3-diazepinone, where it can be seen that they both dock with ASN54 and ALA58 residues.

#### 2.3.5. Inhibition of Proto-Oncogene Tyrosine-Protein Kinase Src

Standard cytotoxic drug dasatinib is an extremely strong inhibitor of the proto-oncogene tyrosine-protein kinase Src (PDB: 3G5D), which is confirmed by our molecular docking simulation where it had a high docking score of −9.4 kcal/mol. Compared to the standard, compound **6** had a moderately high docking score of −7.6 kcal/mol (Table 4). Binding interactions with the active site of the protein revealed that compound **6** formed strong hydrogen bonds with GLY274, GLN275, GLU280, and ALA390 residues as well as other alkyl and pi-alkyl hydrophobic interactions with several residues. Compound **6** shared similar docking sites as dasatinib with several residues of the tyrosine-protein kinase Src, which included LEU273, VAL281, ALA293, CYS345, ALA390, LEU393, and ALA403 (Figure 7).

#### 2.3.6. Inhibition of Human Thymidine Kinase 1

Against the human thymidine kinase 1 protein, compound **6** and the standard ligand thymidine-5′-triphosphate both demonstrated strong binding affinity with a similar docking score of −8.0 kcal/mol (Table 4). Both the standard ligand and compound **6** formed strong hydrogen bonds with LYS32 and ARG60 residues in the active site of human thymidine kinase 1 located at chain H. Additionally, compound **6** also formed strong hydrogen bonds with TYR61 and GLN100 residues and had hydrophobic interactions with PHE29 and ILE45 residues as well (Figure 8).

#### 2.3.7. Inhibition of Human Thymidylate Synthase

Human thymidylate synthase is the target protein of various cytotoxic drugs, which include 5-fluorouracil. In the molecular docking simulation against human thymidylate synthase, compound **6** demonstrated only a moderate docking score (−7.8 kcal/mol) towards the active site in comparison to the docking score demonstrated by 5-fluorodeoxyuridine monophosphate (−9.1 kcal/mol), which is the active metabolite of standard cytotoxic drug 5-fluorouracil (Table 4). The standard 5-fluorodeoxyuridine monophosphate has a total of eleven hydrogen bonds with ARG50, ARG175, ARG176, CYS195, GLN214, ARG215, SER216, GLY217, ASP218, ASN226, and HIS256 residues in the active site of human thymidylate synthase, which makes it a very potent inhibitor. In comparison, compound **6** has very few docking sites similar to that of the standard ligand and only forms hydrogen bonds with ASP174, PRO303, and PRO305 residues of human thymidylate synthase (Figure 9).

#### 2.3.8. Inhibition of Human Adenosine Deaminase 2

Compound **6** had a much higher docking score of −8.0 kcal/mol against human adenosine deaminase 2 protein, in comparison to the −7.4 kcal/mol docking score demonstrated by its potent inhibitor coformycin (Table 4). This strong binding affinity of compound **6** could be attributed to its strong hydrogen formed with HIS301 residue and hydrophobic interactions with ILE117, TRP178, PHE185, PHE186, ARG222, LEU224, and HIS267 residues at the active site of the protein. In comparison, the standard ligand coformycin only formed bonds with ASP89, GLU182, and PHE185 residues of human adenosine deaminase 2 (Figure 10).

## 3. Discussion

Intensive work has been undertaken over the past few decades to develop efficient, target-specific, and safe novel sugar and nucleoside derivatives for the treatment of several diseases [30,31]. Structural modification of the nucleosides’ bases and sugar moiety has led to the development of newer, more potent nucleoside and nucleotide drugs through improving pharmacokinetic properties and target specificity [32]. Even a small change to the structure of nucleoside and nucleotide analogs have profound effects on the biological properties of the compounds. Several nucleoside analogs such as deoxyadenosine analogs, deoxyguanosine, thymidine, and deoxyuridine analogs are extensively used as anticancer, antiviral, antitubercular, immunosuppressant, and HIV reverse transcriptase inhibitors [4,24,33]. There are still numerous sugar derivatives and nucleoside analogs yet to be assessed for their different pharmacological potentials.

Nucleoside analogs have been reported to have analgesic properties in a number of antinociceptive assays [34,35,36]. In our study, compounds **2**, **7,** and **8** all showed a significant reduction in the number of writhes in mice caused by acetic acid, which indicates the peripheral antinociceptive characteristics of the compounds [37]. One possible hypothesis behind this analgesic activity could be the inhibition of adenosine kinase. Adenosine, which is chemically a purine ribonucleoside, is a neurotransmitter present throughout the body and controls a number of bodily functions. Inhibition of the adenosine metabolizing enzyme, adenosine kinase, increases extracellular adenosine concentrations at sites of tissue trauma, and thus adenosine kinase inhibitors may have therapeutic potential as an analgesic and anti-inflammatory agent [38]. Nucleosides have been clinically proven to possess antinociceptive characteristics through inhibition of adenosine kinase [34], though no molecular docking simulations were found in the literature to support this theory of inhibition. Our study is the first ever reporting of the synthesis and analgesic property of compound **8**, as well as the first attempt to find the binding affinity of synthesized nucleosides against adenosine kinase through molecular docking simulation.

To determine the molecular mechanistic pathway of demonstrated analgesic activity, molecular docking simulations were also conducted for the synthesized compounds. Compounds **2**, **7,** and **8** were docked with the actives sites of cyclooxygenase enzymes and compared with a commercially available inhibitor aceclofenac. Inflammatory mediators released during trauma, such as prostaglandins, alter the firing threshold of nociceptors by sometimes directly stimulating them. Hence, inhibition of cyclooxygenase enzymes, particularly cyclooxygenase-2, plays a key role in inducing analgesic activity [39]. All three compounds had a strong binding affinity toward the COX-1 and COX-2 enzymes. The 3D models of the ligand–protein interaction revealed that the compounds bound deeply inside the pocket of cyclooxygenase enzymes through a number of non-covalent hydrogen bonds and pi-alkyl interactions (Figure 3 and Figure 4). These bonds help to intercalate the drug in the binding site of the protein through charge transfer. Molecular docking simulation was also performed against human adenosine kinase and compared with a previously reported potent inhibitor 5-ethynyl-7-(beta-d-ribofuranosyl)-7H-pyrrolo [2,3-d]pyrimidin-4-amine [40]. Interestingly, nucleoside analogs **7** and **8** had strong docking scores against human adenosine kinase and also had a very similar binding site to that of the standard ligand we used in our study (Table 3). The new compound **8** had even more hydrogen bonds and pi-alkyl interactions than the standard ligand and unlike the standard, it did not suffer from any unfavorable bonds with the binding site (Figure 5). This is a very exciting discovery as it not only validates the hypothesized theory of adenosine kinase inhibition by nucleoside analogs, but it also paves the way for new analog development from compound **8,** which will have even better specificity towards the target enzyme, and hopefully better antinociceptive activity as well.

The compounds synthesized in this study were tested for anti-cancer properties on HeLa cell lines, where compound **6** demonstrated significant cytotoxic activity in a dose-dependent manner with an IC_50_ value of 54 µg/mL (Table 2). Nucleoside analogs have been used as chemotherapeutic agents in the management of cancer for years, and several newer analogs are now under clinical trials [41]. A library of thirty-nine bridged nucleosides synthesized by Nicolaou et al. demonstrated potent cytotoxic activity on CCRF-CEM (acute lymphoblastic leukemia derived human T leukemic lymphoblasts) and Raji (Burkitt’s lymphoma derived human B lymphocytes) cell lines [42]. In another study, Marzabadi et al. performed cycloaddition reactions between barbiturate derived thionoimides and glycals in modest yields to synthesize novel bridging nucleoside analogs [43]. The compounds were tested for cytotoxicity on human PBM (peripheral blood mononuclear) cells, CEM cells, and Vero (African green monkey kidney) cells where the most potent compounds demonstrated IC_50_ values ranging from 12.6 μM to up to 100 μM and above. Several other nucleoside analogs have been FDA approved over the last few years and many others are in process of approval as anticancer agents [44]. As our synthesized compound **6** showed an IC_50_ value of 54 µg/mL on the HeLa cell line, it could be used for synthesizing further nucleoside analogs or bridged nucleosides with even more potent chemotherapeutic activity. To our knowledge, this is the first ever cytotoxicity analysis of our synthesized compounds performed on the HeLa cell line.

Chemotherapeutic agents have a multifarious mechanistic pathway and thus have numerous target receptors and proteins. Thus, we chose several target proteins to discern the cytotoxic action demonstrated by compound **6** based on the Swiss Target Prediction online tool and performed a molecular docking simulation against these target proteins. Human cytidine deaminase, which was one of our selected protein targets, is a ubiquitous enzyme essential to DNA and RNA synthesis due to its recycling role of free pyrimidines in the pyrimidine salvage pathway [45]. Overexpression of cytidine deaminase has been associated with chemotherapy-related resistance to treatment, due to its catalytic role in the metabolic processing of nucleoside type anticancer and antiviral agents. Therefore, potent inhibitors of human cytidine deaminase such as 1-beta-ribofuranosyl-1,3-diazepinone are potential treatment strategies in cancer patients as small molecule therapeutic adjuvants [46]. In our molecular docking simulation, we found compound **6** to have a better docking score (−7.1 kcal/mol) than the standard ligand 1-beta-ribofuranosyl-1,3-diazepinone (−6.8 kcal/mol) (Table 4). This is probably due to the higher number of active site residues that compound **6** is interacting with, compared to the standard ligand (Figure 6). This finding is significant because human cytidine deaminase inhibitory action would be an added benefit for any potential cytotoxic compound and it might help to suppress cancer chemoresistance [45].

The Src family of non-receptor protein tyrosine kinases regulate a number of cellular processes such as cell division, survival, motility, angiogenesis, etc., due to their role in important cellular signal transduction pathways. Overexpression of proto-oncogene tyrosine-protein kinase Src, also known as c-Src, is often aberrantly activated or overexpressed in several epithelial and non-epithelial cancers [47]. Therefore, c-Src is a potential therapeutic target in cancer patients with solid tumors. Dasatinib is a potent small molecule inhibitor of tyrosine-protein kinase Src and is a targeted chemotherapeutical medication used in the treatment of neoplasia including chronic myelogenous leukemia and acute lymphoblastic leukemia [48]. Molecular docking against tyrosine-protein kinase Src revealed that compound **6** had comparatively lesser docking scores (−7.6 kcal/mol) than the standard drug dasatinib (−9.4 kcal/mol). Even though it had several hydrogen bonds and pi-alkyl interactions formed with the active site of tyrosine-protein kinase Src, compound **6** probably did not bind into the deep pockets of the protein (Figure 7).

Compound **6** had an excellent binding affinity towards the cytoplasmic human thymidine kinase 1 protein. The cell-cycle regulated cytoplasmic human thymidine kinase 1 has an important role in cell cycle modulation due to its catalytic role in the phosphorylation of thymidine in the salvage pathway [49]. Dysregulation of human thymidine kinase 1 has been reported to have an association with the progression of several human malignancies [50]. Thymidine-5′-triphosphate is a strong feedback inhibitor of human thymidine kinase-1 and is a native ligand of the enzyme [51]. Molecular docking simulations revealed that synthesized compound **6** had exactly the same docking score (−8.0 kcal/mol) as the standard ligand thymidine-5′-triphosphate (Table 4). This high binding affinity of compound **6** could be attributed to its interaction with the active site of the protein through several hydrogen and pi-alkyl bonds (Figure 8).

The thymidylate synthase enzyme is a critical target in cancer chemotherapy because of its crucial role in the synthesis of 2′-deoxythymidine-5′-monophosphate, which is a necessary precursor for DNA biosynthesis [52]. 5-fluorouracil is an antimetabolite drug, which is the first potent inhibitor of thymidylate synthase. In the body, 5-fluorouracil is converted to its active metabolite 5-fluorodeoxyuridine monophosphate, which inhibits thymidylate synthase, hence the drug is widely used for treating colorectal, pancreatic, ovarian, breast, etc., cancer patients [52]. We conducted molecular docking of our synthesized compounds and standard 5-fluorodeoxyuridine monophosphate against the human thymidylate synthase enzyme. Our analysis revealed that cytotoxic compound **6** had a much comparatively lesser docking score (−7.8 kcal/mol) than the standard ligand (−9.1 kcal/mol) (Table 4). The 3D representation of the binding interactions shows that compound **6** binds with the active site via a much lesser number of hydrogen bonds compared to 5-fluorodeoxyuridine monophosphate (Figure 9). Hence, it cannot be considered a potent inhibitor of the human thymidylate synthase enzyme.

Adenosine is an important purine that regulates several physiological functions by interacting with adenosine receptors located in different regions of the body. Intra- and extracellular concentration of adenosine is regulated by the adenosine deaminase enzyme, which converts adenosine to inosine. Overexpression of adenosine deaminase has been reported in the progress of various diseases, including cancers, which is why inhibition of the enzyme may be a possible treatment strategy [53]. Coformycin is a strong nucleoside inhibitor of adenosine deaminase and is considered to have potent anticancer properties [54]. Synthesized compound **6** had a much higher docking (−8.0 kcal/mol) score against human adenosine deaminase type 2 than the standard drug coformycin (−7.4 kcal/mol) (Table 4). This higher docking score could be attributed to the higher number of hydrogen bond and hydrophobic bonds such as pi-cation, pi-pi stacked, pi-pi T-shaped, alkyl and pi-alkyl interactions of compound **6** with active site of human adenosine deaminase 2, compared to coformycin (Figure 10).

## 4. Materials and Methods

### 4.1. Chemicals and Reagents

All solvents and reagents required for synthesis were collected from commercial sources. In some cases, solvents were purified and dried by refluxing with CaH_2_ followed by distillation. Clean and well-dried glassware was used for reactions. The reactions which were sensitive to moisture were carried out under N_2_ gas stream using well-dried glassware. PSQ-60B or PSQ-100B silica gel was used for column chromatography. Analytical thin-layer chromatography (TLC) on aluminum plates (TLC silica gel 60 F254) was used to monitor the progress of the reactions where the products were either visualized by UV light at 254 nm or staining with a solution of 1:1:18 (*v*/*v*) p-anisaldehyde/H_2_SO_4_/ethanol and subsequent charring with a heat gun.

### 4.2. Synthesis

The chemical synthesis of several sugar and nucleoside analogs (**2**–**8**) started from α-d-glucose (**1**) according to Figure 1. A number of chemical steps were involved in the structural modification of the starting compound **1** to synthesize several sugar and nucleoside derivatives.

#### 4.2.1. Preparation of 1,2:5,6-Di-*O*-Isopropylidene-α-d-Allofuranose (**2**)

Synthesis of compound **2** from α-d-glucose (**1**) was carried out involving three steps following reported methods [21,22,23] and has been described in the Appendix A.

Yield: 4.12 g (38% in three steps), ^1^H-NMR (301 MHz, CDCl_3_) δ_H_ (ppm) = 5.82 (d, *J* = 3.8 Hz, 1H), 4.62 (t, *J* = 4.5 Hz, 1H), 4.33 (dd, *J* = 6.5, 4.8 Hz, 1H), 4.11–3.99 (m, 3H), 3.82 (dd, *J* = 8.6, 4.8 Hz, 1H), 2.54 (d, *J* = 8.3 Hz, 1H), 1.58 (s, 3H), 1.47 (s, 3H), 1.38 (s, 6H).

#### 4.2.2. Preparation of 3-*O*-Benzyl-4-*C*-Hydroxymethyl-1,2-*O*-Isopropylidene-α-d-Ribofuranose (**3**)

Synthesis of compound **3** from compound **2** was accomplished in four steps according to reported methods [24,25] and has been described in the Appendix A.

Yield: 1.37 g (53%), ^1^H-NMR (500 MHz, CDCl_3_) δ_H_ (ppm) = 7.34 (m, 5H), 5.77 (d, *J* = 2.4 Hz, 1H), 4.81 (d, *J* = 7.2 Hz, 1H), 4.65 (t, *J* = 1.9 Hz, 1H), 4.56 (d, *J* = 6.9 Hz, 1H), 4.22 (d, *J* = 3 Hz, 1H), 3.91 (d, *J* = 4.5 Hz, 2H), 3.79 (dd, *J* = 2.4, 7.2 Hz, 1H), 3.57 (t, *J* = 4.3 Hz, 1H), 2.35 (t, *J* = 4.2 Hz, 1H), 1.80 (dd, *J* = 2.1, 6.0 Hz, 1H), 1.63 (s, 3H), 1.56 (s, 3H).

#### 4.2.3. Preparation of 1-(4-*C*-Acetoxymethyl-2-*O*-Acetyl-3,5-Di-*O*-Benzyl-β-d-Ribofuranosyl)Thymine (**4**)

Synthesis of nucleoside analog **4** was conducted from compound **3** in three steps based on methods reported in the literature with slight modifications [26,27,28] and has been described in the Appendix A.

Yield: 3.34 g (63%), ^1^H-NMR (301 MHz, CDCl_3_) δ_H_ (ppm) = 7.96 (s, 1H), 7.40–7.26 (m, 11H), 6.22 (d, *J* = 5.5 Hz, 1H), 5.41 (t, *J* = 5.5 Hz, 1H), 4.60 (t, *J* = 11.2 Hz, 1H), 4.52–4.40 (m, 4H), 4.19–4.09 (m, 2H), 3.77 (d, *J* = 10.1 Hz, 1H), 3.51 (d, *J* = 10.1 Hz, 1H), 2.10 (s, 3H), 2.04 (s, 3H), 1.52 (s, 3H).

#### 4.2.4. Preparation of 1-(4-*C*-Hydroxymethyl-2-Hydroxyl-3,5-Di-*O*-Benzyl-β-d-Ribofuranosyl)Thymine (**5**)

Nucleoside analog 5 was synthesized from compound **4** according to the reported literature [28], which has been described in the Appendix A.

Yield: 1.6 g (84%), ^1^H-NMR (301 MHz, CD_3_OD) δ_H_ (ppm) = 7.9 (s, 1H), 7.61 (s, 1H), 7.6–7.27 (m, 10H), 6.03 (d, *J* = 4.4 Hz, 1H), 4.8 (s, 1H), 4.55 (s, 1H), 4.5 (d, *J* = 5.4, 2H), 4.37 (dd, 4.5, 5.7 Hz, 1H), 4.28 (d, *J* = 5.7 Hz, 1H), 4.07 (dd, *J* = 7.2, 14.4 Hz, 1H), 3.57 (d, *J* = 2.7 Hz, 2H), 3.53 (s, 2H), 2.01 (s, 1H), 1.43 (s, 3H), 1.21 (t, *J* = 6.9 Hz, 1H).

#### 4.2.5. Preparation of 3′,5′-Di-*O*-Benzyl-4′-*C*-Tert-Butyldiphenylsiloxymethyl-5-Methyluridine (**6**)

Compound **6** was produced from compound **3** via four steps according to reported methods [27,28] and has been described in the Appendix A.

Yield: 9.3 g (53%), ^1^H-NMR (301 MHz, CDCl_3_) δ_H_ (ppm) = 7.97 (s, 1H), 7.67–7.60 (m, 5H), 7.47–7.21 (m, 16H), 5.95 (d, *J* = 4.8 Hz, 1H), 4.70 (dd, *J* = 11.0, 19.6 Hz, 2H), 4.50 (s, 2H), 4.39 (dd, *J* = 4.8, 11.1 Hz, 1H), 4.29 (d, *J* = 6.0 Hz, 1H), 3.76 (dd, *J* = 10.8, 28.5 Hz, 2H), 3.6 (dd, *J* = 8.7, 10.5 Hz, 2H), 3.5 (s, 1H), 1.59 (s, 3H), 1.06 (s, 9H).

#### 4.2.6. Preparation of (1S,3R,4R,7S)-7-Hydroxy-1-Hydroxymethyl-3-(Thymin-1-Yl)-2,5-Dioxabicyclo [2.2.1]Heptane (**7**)

Bridged nucleoside **7** was produced from compound **3** in six steps according to previously reported methods [9,28,29], and has been described in the Appendix A.

Yield: 1.57 g (97%), ^1^H-NMR (301 MHz, DMSO-D_6_) δ_H_ (ppm) = 11.33 (s, 1H), 7.61 (s, 1H), 5.63 (d, *J* = 4.5 Hz, 1H), 5.4 (s, 1H), 5.17 (t, *J* = 5.7 Hz, 1H), 4.1 (s, 1H), 3.9 (d, *J* = 4.1 Hz, 1H), 3.81 (d, *J* = 7.6 Hz, 1H), 3.75 (d, *J* = 5.5 Hz, 2H), 3.62 (d, *J* = 7.6 Hz, 1H), 1.77 (s, 3H).

#### 4.2.7. Preparation of ((1R,3R,4R,7S)-7-Hydroxy-3-(5-Methyl-2,4-Dioxo-3,4-Dihydropyrimidin-1(2H)-yl)-2,5-Dioxabicyclo [2.2.1]Heptan-1-yl)Methyl Isobutyrate (**8**)

Compound **7** (0.6 g, 2.22 mmol) was dissolved in pyridine (23 mL, 285.53 mmol) and stirred at room temperature under N_2_ gas until everything dissolved. Isobutyric anhydride (0.55 mL, 3.31 mmol) was added to the reaction mixture and stirred for 24 h during which reaction progression was monitored by TLC using a chloroform-methanol (9:1 *v*/*v*) solvent system. After the reaction was complete, residual pyridine was dried out by co-evaporating with toluene. This co-evaporation step was performed several times to ensure that all pyridine had been removed from reaction medium, otherwise, the residual pyridine interferes with the column chromatography by easily passing through the column and mixing with product. Finally, the concentrated crude was purified by column chromatography using chloroform-methanol (9.5:0.5 *v*/*v*) solvent system to produce solid crystalline compound **8** (Figure 1).

Yield: 0.42g (56%), R_f_ = 0.7 (9:1 (*v*/*v*) CHCl_3_/CH_3_OH.); IR (KBr, cm^−1^): 3258.97 (N-H), 3164.55 (O-H), 2972.85 (C-H stretch), 1736.78 (C=O), 1093.0 (C-O), 1051.51 (C-N). ^1^H NMR (400 MHz, CDCl_3_) δ_H_ (ppm) = 8.30 (s, 1H), 7.39 (s, 1H), 5.61 (s, 1H), 4.63 (d, *J* = 12.8 Hz, 1H), 4.52 (s, 1H), 4.35 (d, *J* = 12.8 Hz, 1H), 4.08 (d, *J* = 8.3 Hz, 1H), 3.90 (d, *J =* 8.3 Hz, 2H), 2.85 (d, *J* = 5.0 Hz, 1H), 2.68 (m, 1H), 1.94 (s, 3H), 1.25 (m, 6H). ^13^C NMR (126 MHz, CD_3_OD) δ: 177.83, 166.36, 151.76, 136.15, 110.87, 88.55, 87.91, 80.81, 72.50, 71.16, 60.53, 35.13, 19.39, 12.68. HRMS calc. for C_15_H_20_N_2_O_7_Na [M + Na]^+^: 363.1160; found: 363.1163.

### 4.3. Biological Study

The synthesized compounds were tested for in vivo peripheral analgesia and in vitro anticancer activities in Swiss albino mice and human cancer HeLa cell line respectively.

#### 4.3.1. In Vivo Peripheral Analgesic Activity

Analgesic activity was evaluated by the acetic acid-induced writhing method in Swiss albino mice [55]. Swiss albino mice (*Mus musculus*) weighing 25–30 g of either sex were obtained from Jahangir Nagar University Animal House. The mice were fed with standard diet and water ad libitum. From 12 h prior to the experiment only water was fed to the animals. Each sample was tested at both 25 mg/kg and 50 mg/kg body weight. Aceclofenac was also used as the positive control at 25 mg/kg body weight. The mice were weighed accurately and randomly divided into 16 groups taking 5 mice in each group. The negative control group was orally given 0.9% normal saline, positive control group was given standard aceclofenac and the other groups were given test solutions (25 and 50 mg/kg doses for each tested compound). Acetic acid (0.7%) was administered intraperitoneally to each mouse after 30 min. Finally, after five minutes, number of squirms or writhing was counted for each mouse for up to fifteen minutes.

The experimental protocols were approved by the ethical committee of the Faculty of Pharmacy, University of Dhaka. Handling of animals was carried out following the rules provided by the ethical committee and all efforts were made to minimize animal suffering. The least number of animals required was used and the intensity of the noxious stimuli possible to demonstrate consistent effects of the drug treatments was kept as low as possible.

#### 4.3.2. In Vitro Cytotoxic Activity

In this study, the cytotoxicity of the synthesized sugar and nucleoside analogs was assessed by observing the morphological analysis of HeLa cells that indicate cell death (such as cell rupture, leakage, etc.), using a microscope [56,57]. The assay was segmented into two phases. In phase one, percentage of cell viability was determined using the trypan blue dye exclusion technique. If any compound showed excellent cytotoxicity in phase one, then it was subjected to quantitative analysis to determine its IC_50_ value (50% growth inhibitory concentration) using the Cell Counting Kit-8 (CCK-8) assay.

#### 4.3.3. Trypan Blue Dye Exclusion Method

The preliminary cytotoxic potential of the synthesized compounds was tested using the trypan blue dye exclusion test which is based on the principle that viable cells present in a cell suspension possess intact cell membranes which make them impermeable to the trypan blue dye [58]. Therefore, any dead cells produced due to administration of cytotoxic compounds will easily take up the trypan blue dye and be stained blue, which can be easily distinguished from the clear cytoplasm of a viable living cell under a microscope.

A fixed volume of cells, (e.g., 1 mL) was used to prepare a cell suspension and 50 µL of this cell suspension was taken in a cryovial. Equal amount of 0.4% trypan blue was also added to the cell suspension and mixed well. A hemocytometer was then used to count the number of living and dead cells in the suspension. Each side of the hemocytometer usually takes about 10–20 µL of the suspension. An inverted microscope was then used to count the number of living cells in the hemocytometer. Living healthy cells will prevent the dye from penetrating inside because of its intact cell membrane and will appear to be clear. On the other hand, dead cells will stain blue because of taking up the dye and become blue in color. Both clear and blue cells in each large square at every corner of the hemocytometer were counted. Percentage of viable cells was counted using the following formula:*Percentage (%) Cell Viability* = (Live cell count/Total cell count) × 100

#### 4.3.4. In Vitro Cytotoxic Activity

Cell Counting Kit-8, a non-radioactive colorimetric cell proliferation and cytotoxic assay kit were used for the cytotoxicity assay [59]. HeLa cells were cultured in DMEM (Dulbecco’s Modified Eagles’ Medium) containing 0.2% gentamycin, 1% penicillin-streptomycin (1:1), and 10% fetal bovine Serum (FBS). A 96-well plate was used to seed the HeLa Cells (2 × 10^4^/100 µL) and then they were incubated at 37 °C under the presence of carbon dioxide for 24 h. The next day, 25 µL of filtered sample was added into each well in duplicate. For control group, only 2.5% DMSO solution was added to the wells seeded with HeLa cells. Compounds that demonstrated cytotoxic potential in the trypan blue dye exclusion method were prepared at a concentration of 250 µg/mL using 2.5% DMSO and serially diluted to prepare 125, 62.5, 31.25 and 15.62 µg/mL solutions. For an incubation period of 48 h, the cells were regularly checked for shrinkage, swelling, and granularity using trinocular microscope fitted with camera. After 48 h, 10 µL of 5 mg/mL CCK-8 solution was added to each well and incubated for a further 4 h at 37° to measure cytotoxicity. A purple-colored formazan dye was formed in viable cells which could be easily visualized. Amount of produced formazan dye is proportional to the number of viable cells and could be measured using a microplate reader at 570 nm wavelength using DMSO as blank. Following formula was used to calculate percentage growth inhibition:*% Cell Inhibition* = [(A_t_ − A_b_)/(A_c_ − A_b_)] × 100

Here, A_t_ = Mean absorbance value of test compound, A_b_ = Mean absorbance value of blank, A_c_ = Mean absorbance value of control.

IC_50_ value is determined by plotting percentage of inhibition against the drug concentration.

### 4.4. In Silico Molecular Docking Simulation

The synthesized sugar derivatives and nucleoside analogs were energy and geometry optimized and docked via computational techniques against selected protein targets to predict their cytotoxic and analgesic potential. Molecular docking studies were performed using computational analysis software packages such as Avogadro (Version 1.2), PyMOL (Version 2.4.1, Schrödinger, Inc., New York, NY, USA), Swiss-PdbViewer (Version 4.1, Swiss Institute of Bioinformatics, Basel, Switzerland), PyRx (Version 0.8, The Scripps Research Institute, San Diego, CA, USA), and Biovia Discovery Studio Visualizer 2016 (Version 16.1, Dassault Systèmes, Vélizy-Villacoublay, France).

#### 4.4.1. Target Protein Selection

To understand the molecular mechanism of cytotoxic and anti-nociceptive action demonstrated by the synthesized compounds, we retrieved three-dimensional (3D) structures of selected proteins from the protein data bank (https://www.rcsb.org) (accessed on 1 February 2022). For studying cytotoxic properties, five protein targets were chosen based on analysis of Swiss Target Prediction online tool (http://www.swisstargetprediction.ch/) (accessed on 1 February 2022). The selected proteins were human cytidine deaminase (PDB: 1MQ0), proto-oncogene tyrosine-protein kinase Src (PDB: 3G5D), human thymidine kinase 1 (PDB: 1XBT), human thymidylate synthase (PDB: 6ZXO) and human adenosine deaminase 2 (PDB: 3LGG). For predicting analgesic activity, cyclooxygenase-1 (PDB: 1EQG), cyclooxygenase-2 (PDB: 5IKT) enzymes, and human adenosine kinase (PDB: 4O1L) were selected. The target proteins were downloaded from the protein data bank in PDB format and prepared for docking by removing all water molecules, heteroatoms, and unwanted ligands using PyMOL 2.4.1. Non-polar hydrogen atoms were added to the proteins and then energy minimization was performed using Swiss-PdbViewer version 4.1 where in vacuo computations were performed without any reaction field using the GROMOS 96 43B1 parameters set.

#### 4.4.2. Ligand Preparation

Initial three-dimensional (3D) structures of the synthesized compounds were obtained from PubChem (https://pubchem.ncbi.nlm.nih.gov/) (accessed on 1 February 2022) if available or drawn using Avogadro 1.2. Standards selected for the target proteins, which included aceclofenac (PubChem CID 71771), 5-ethynyl-7-(beta-D-ribofuranosyl)-7H-pyrrolo [2,3-d]pyrimidin-4-amine (PubChem CID 49784135). 1-beta-ribofuranosyl-1,3-diazepinone (PubChem CID 5287841), dasatinib (PubChem CID 3062316), thymidine-5′-triphosphate (PubChem CID 64968), 5-fluorodeoxyuridine monophosphate (PubChem CID 8642) and coformycin (PubChem CID 25447) were obtained from PubChem. The structures of compounds and standards were then geometrically and energy optimized, followed by vibrational frequency calculation, using Hartree–Fock (HF) level of theory with the standard STO-3G basis set [60].

#### 4.4.3. Ligand–Protein Interaction

Molecular docking simulations were performed using PyRx 0.8 for predicting the binding interactions of synthesized compounds with target proteins and comparing the ligand–protein interactions with established drugs and previously reported potent inhibitors. Binding affinities during drug–protein linking process were determined using the semiflexible modeling feature of AutoDock Vina. Proteins were loaded and formatted as target macromolecule in PyRx while the compounds were imported and formatted as ligands. Open Babel tool of AutoDock Vina was used to minimize the ligands into pdbqt format in order to fit the best optimal hit during docking against the active site of target proteins. Active site coordinates of the proteins were obtained using PyMOL and those coordinates were then used to center the position of the grid box generated in PyRx. All other features were kept in default settings and molecular docking simulations were performed using the AutoDock Vina feature of PyRx. Best ligand–protein interactions were analyzed from the 2D and 3D docking positions of the compounds using BIOVIA Discovery Studio Visualizer 2016.

### 4.5. Statistical Analysis

Statistical analysis was performed using SPSS (Version 25.0, IBM, New York, NY, USA) and Microsoft Excel 2019 (Microsoft, Washington, USA). Values are presented as the mean ± standard error of mean (SEM). One-way analysis of variance (ANOVA) followed by post hoc Dunnett’s test was carried out where *p* < 0.05 was considered statistically significant, respectively.

## 5. Conclusions

In this study, several sugar and nucleoside analogs were synthesized from α-d-glucose, and their purity was confirmed by spectroscopic methods. The final compound **8** was obtained for the first time in 21 steps. Seven synthesized compounds were subjected to biological assays such as in vivo analgesic activity and in vitro cytotoxicity that had not been explored before. Compounds **2**, **7,** and **8** demonstrated significant peripheral anti-nociceptive characteristics in the acetic acid-induced writhing model. Molecular docking simulations suggest that compounds **2**, **7,** and **8** may be potent inhibitors of cyclooxygenase-2 and human adenosine kinase enzymes. On the other hand, compound **6** showed significant cytotoxic properties with an IC_50_ value of 54 µg/mL, which could be attributed to its potent inhibition of human cytidine deaminase, human thymidine kinase, and human adenosine deaminase enzymes. This is the first report for evaluating experimental and computational analgesic and cytotoxic effects of the synthesized sugar and nucleoside derivatives. Further structural modifications based on binding interactions demonstrated by the compounds will lead to the development of newer derivatives with even more potent biological properties.

## Data Availability

Data is contained within the article and Appendix A.

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
