# Peer review of "Synthesis of Sugar and Nucleoside Analogs and Evaluation of Their Anticancer and Analgesic Potentials"

_molecules, 2022, doi:10.3390/molecules27113499_

Round 1

Reviewer 1 Report

The paper describes the synthesis of several sugar-derived nucleosides and the study of their anticancer and analgesic activity.
It is correct in its synthesis and activity studies and perhaps most notable in its attempt to rationalize the observed activities with the molecules structure.
It can be published as presented, but it can also be rewritten to focus more on the results and thus make it more interesting for the general scientific community.
Much of the information is not of general interest and can be included in supporting information, as it may be of interest to those who want to go deeper into the work.
For example: it is of no interest to describe the synthesis of compound 2, as it is commercial and cheap. Besides, its preparation only requires the conditions ia.

Reviewer 2 Report

1. In table 3, the binding affinities of 2, 7, 8 with 1EQG, 5IKT, 4O1L are very similar, can the author explain why? Or have the authors tried docking with some control proteins that will not bind with these ligands?

2. Synthesis of scheme 1, please refer to Chemical Biology & Drug Design, 2015, 85(3), 245-252.

3. The anti-cancer studies here seem a little bit distracted since most of the studies focus on analgesic activity study, the author might want to remove that part so the article flows better.
